# Brain Structural Connectivity Differences in Patients with Normal Cognition and Cognitive Impairment

**DOI:** 10.3390/brainsci11070943

**Published:** 2021-07-18

**Authors:** Nauris Zdanovskis, Ardis Platkājis, Andrejs Kostiks, Guntis Karelis, Oļesja Grigorjeva

**Affiliations:** 1Department of Radiology, Riga Stradins University, Dzirciema Street 16, LV-1007 Riga, Latvia; ardis.platkajis@rsu.lv; 2Department of Radiology, Riga East University Hospital, Hipokrata Street 2, LV-1038 Riga, Latvia; 3Department of Neurosurgery and Neurology, Riga East University Hospital, Hipokrata Street 2, LV-1038 Riga, Latvia; andrejs.kostiks@gmail.com (A.K.); guntis.karelis@gmail.com (G.K.); 4Department of Computer Control Systems, Riga Technical University, Kaļķu Street 1, LV-1658 Riga, Latvia; Olesja.Grigorjeva@rtu.lv

**Keywords:** DTI, MRI, brain connectivity, MCI, dementia, mild cognitive impairment, neurodegenerative diseases

## Abstract

Advances in magnetic resonance imaging, particularly diffusion imaging, have allowed researchers to analyze brain connectivity. Identification of structural connectivity differences between patients with normal cognition, cognitive impairment, and dementia could lead to new biomarker discoveries that could improve dementia diagnostics. In our study, we analyzed 22 patients (11 control group patients, 11 dementia group patients) that underwent 3T MRI diffusion tensor imaging (DTI) scans and the Montreal Cognitive Assessment (MoCA) test. We reconstructed DTI images and used the Desikan–Killiany–Tourville cortical parcellation atlas. The connectivity matrix was calculated, and graph theoretical analysis was conducted using DSI Studio. We found statistically significant differences between groups in the graph density, network characteristic path length, small-worldness, global efficiency, and rich club organization. We did not find statistically significant differences between groups in the average clustering coefficient and the assortativity coefficient. These statistically significant graph theory measures could potentially be used as quantitative biomarkers in cognitive impairment and dementia diagnostics.

## 1. Introduction

In recent years, scientists have been able to study the brain in ways they never could before, due to advancements in imaging technologies. One of these advances is diffusion tensor imaging (DTI), which tracks water movement through the brain’s white matter [1]. DTI has allowed researchers to analyze DTI metrics and brain connectivity differences between patients with cognitive impairment and normal cognition [2,3]. Brain connectivity plays an important role in sustaining the function of the human brain and cognition. Thus, analyzing brain connectivity could potentially lead us to imaging biomarker discoveries that could aid in cognitive impairment diagnostics [4,5].

It is important to distinguish between functional and structural connectivity. Functional connectivity can be tracked, measured, and analyzed using functional methods, i.e., functional magnetic resonance imaging (fMRI), positron emission tomography (PET), electroencephalography (EEG), magnetoencephalography (MEG), and others [6,7].

On the other hand, white matter bundles and tracts can be visualized by utilizing diffusion imaging on magnetic resonance machines, particularly diffusion tensor imaging (DTI). By utilizing DTI, it is possible to visualize white matter tracts, reconstruct them, and apply graph theory measures to analyze differences in these tracts among individuals [8]. These measurements can be used in specific brain regions (specific white matter tracts) or could be used in a whole-brain connectometry analysis [9].

To perform whole-brain connectometry analysis, it is necessary to define which cortical atlas would be used for brain parcellation. Several available cortical parcellation atlases are used in neuroimaging and network analysis, i.e., the Desikan–Killiany–Tourville (DKT) atlas [10], the automated anatomical labeling 3 (AAL3) atlas [11], Brainnetome Atlas [12], and others. In our study, we used DKT cortical parcellation.

In graph theory, numerous measures could be used as a quantitative variable to assess brain connectivity patterns, for example, ***measures of integration*** (degree level, shortest path length, number of triangles, characteristic path length, global efficiency, clustering coefficient, transitivity, local efficiency, modularity), ***measures of centrality*** (closeness centrality, betweenness centrality, within-module degree, participation coefficient), ***network motifs*** (anatomical and functional motifs, motif z-score, motif fingerprint), ***measures of resilience*** (degree distribution, average neighbor degree, assortativity coefficient), and other measures [13].

In our study, we analyzed undirected and unweighted graph relationships and included:*Graph density* (the mean network degree);*Average clustering coefficient* (fraction of triangles around an individual node);*Network characteristic path length* (the average shortest path length between all pairs of nodes in the network);*Small-worldness* (small-world networks are significantly more clustered than random networks, yet have approximately the same characteristic path length as random networks);*Global efficiency* (average inverse shortest path length);*Assortativity coefficient* (correlation coefficient between the degrees of all nodes on two opposite ends of a link);*Rich club coefficient*, k = 5; 10; 15; 20 (network high-degree nodes that, on average, are more intensely interconnected than lower-degree nodes).

The goal of this research was to identify the changes in structural connectivity between the control group and the dementia group.

## 2. Materials and Methods

### 2.1. Participants

Participants were admitted to a neurologist with suspected cognitive impairment. All participants were evaluated by a board-certified neurologist, and the Montreal Cognitive Assessment (MoCA) was performed.

All participants in the control group and dementia group had at least 16 years of education. For the control group, we used MoCA cutoff scores of ≥23, and for the dementia group, we used a cutoff of ≤22, as these values can produce a high sensitivity, specificity, and AUC [14,15].

Based on the neurological assessment and MoCA test results, patients were divided into two groups: the control group (patients with no severe cognitive impairment), and the dementia group (patients with cognitive impairment who need supervision or some help during daily activities).

The patient demographic data and MoCA scores are shown in Table 1.

Exclusion criteria for study patients were clinically significant neurological diseases (tumors, major stroke, malformations, etc.), drug use, and alcohol abuse. All study patients did not have any other significant abnormalities on magnetic resonance (MR) scans.

Based on the neurological assessment and MRI scans, all patients had a mixed type of dementia.

### 2.2. Magnetic Resonance Imaging (MRI) Data Acquisition and Image Reconstruction

MRI was performed at a single-site university hospital to avoid inter-scanner differences. All scans were converted from DICOM format to Neuroimaging Informatics Technology Initiative (NIfTI) format, and further to SRC and FIB format to perform tractography analysis.

The diffusion images were acquired on a GE SIGNA Architect 3T scanner using a diffusion sequence (TE = 101.5 ms, TR = 14,884 ms). A DTI diffusion scheme was used, and a total of 30 diffusion sampling directions were acquired. The b-value was 1000.59 s/mm^2^. The in-plane resolution was 0.9375 mm. The slice thickness was 2 mm. The b-table was checked by an automatic quality control routine to ensure its accuracy [16]. The b-table was flipped by 0.012 fz. The restricted diffusion was quantified using restricted diffusion imaging [17]. The diffusion data were reconstructed using generalized q-sampling imaging [18], with a diffusion sampling length ratio of 1.25. A deterministic fiber tracking algorithm [19] was used with augmented tracking strategies [20] to improve reproducibility. A seeding region was placed at the whole brain. The anisotropy threshold was randomly selected. The angular threshold was randomly selected from 15 degrees to 90 degrees. The step size was randomly selected from 0.5 to 1.5 voxels. Tracks with a length shorter than 30 or longer than 300 mm were discarded. A total of 10,000,000 seeds were placed.

Free Surfer DKT was used as the brain parcellation, and the connectivity matrix was calculated by using the count of the connecting tracks. The connectivity matrix and graph theoretical analyses were conducted using DSI Studio (http://dsi-studio.labsolver.org, accessed on 3 may 2021) [21].

Acquired images after tractography reconstruction, fiber tracking, and connectivity matrix analysis are shown in Figure 1.

### 2.3. Statistical Analysis

Data were analyzed using statistical analysis software JASP Version 0.14.1. The Mann–Whitney U test was used to determine statistically significant differences between the control group and the dementia group. Results with a *p*-value smaller than 0.05 were considered statistically significant. Descriptive statistics for statistically significant network measures were calculated.

## 3. Results

In total, 22 patients were included in this study: 11 patients in the control group (7 females, 4 males; average age 62.2 years, standard deviation 15.4, median age 69, minimum 35, maximum 77), and 11 patients in the dementia group (7 females, 4 males, average age 75.0 years, standard deviation 10.5, median age 71, minimum 65, maximum 96).

The Mann–Whitney U test results between study groups are shown in Table 2.

We found statistically significant differences between groups in the graph density, network characteristic path length, small-worldness, global efficiency, and rich club coefficient (k = 5; 10; 15; 20). We did not find statistically significant differences between groups in the average clustering coefficient and the assortativity coefficient. Additionally, we analyzed descriptive statistics for statistically significant measures.

### 3.1. Graph Density

The graph density is the fraction of present connections to possible connections [13]. Our findings on the graph density are shown in Table 3. The graph density was higher in control group patients.

### 3.2. Average Clustering Coefficient

The clustering coefficient represents the degree to which nodes in a graph tend to cluster together [22]. Our findings on the average clustering coefficient are shown in Table 4. The average clustering coefficient was higher in the control group patients.

### 3.3. Network Characteristic Path Length

The network characteristic path length (or average path length) is defined as the average distance between all pairs of vertices [23]. Our findings on the network characteristic path length are shown in Table 5. The network characteristic path length was shorter in control group patients.

### 3.4. Small-Worldness

A small-world network is a graph where most nodes are not neighbors, but the neighbors of any given node are likely to be neighbors of each other, and most nodes can be reached from every other node by a small number of edges [24]. Our findings on the small-worldness coefficient are shown in Table 6. The small-worldness coefficient was higher in control group patients.

### 3.5. Global Efficiency

Global efficiency is defined as the average inverse shortest path length in the network [25]. Our findings on global efficiency are shown in Table 7. Global efficiency was higher in control group patients.

### 3.6. Rich Club

Rich club regions are a set of highly interconnected nodes forming a tight subnetwork within a network. In rich club analysis, *k* represents the number of connections attached to each network node [26]. In our analysis, we used k values of 5, 10, 15, and 20. In all cases, there were statistically significant differences between groups, and higher results were found in control group patients. Our findings on rich club organization are shown in Table 8.

## 4. Discussion

There are different approaches on how to analyze structural connectivity in brain. It is possible to analyze whole-brain connectivity patterns or focus on specific brain regions and DTI metrics (fractional anisotropy, mean diffusivity, number of streamlines etc.) [27,28]. In our study, we performed whole-brain connectometry analysis and compared graph theory measures between patients with normal cognition and patients with cognitive impairment. We used the DKT cortical parcellation atlas to estimate graph theory values and quantitative measurements.

In general, connectometry analysis in patients with cognitive impairment could be a promising diagnostic method that provides quantitative biomarkers [29].

Our results on ***global efficiency*** are consistent with other studies that compared brain network global efficiency and cognitive abilities, i.e., global efficiency was a significant predictor of working memory performance [25]. Lower global efficiency values are a significant predictor factor in conversion to dementia in patients with small vessel disease [30].

In our study, the ***characteristic path length*** was higher in patients with cognitive impairment, which is consistent with Alzheimer’s dementia (AD)-related changes [31], and was found to be consistent with a functional network study where a longer path length was observed in AD patients [32].

In our study, the ***small-worldness coefficient*** was lower in patients with cognitive impairment. There are reports where patients with AD tend to lose small-worldness properties [31,32,33].

There were statistically significant differences in ***rich club organization*** with all measured coefficients (k = 5, 10, 15, 20), with higher values in the control group. Rich club organization is very important for global brain communication and global integration of information [34,35]. Rich club disruption could be connected to early-onset AD [36], and, in general, it is associated with MCI and AD-related changes in the brain [37,38].

Although we did not find statistically significant differences between groups in the ***clustering coefficient***, in other research papers, it was stated that in patients with AD, the clustering coefficient is higher than in control subjects [31].

### Limitations

This was an exploratory research study with a limited patient cohort. To confirm the theses stated in this article, we intend to continue the research and validate the results on a larger patient cohort.

## 5. Conclusions

In our study, the graph density, network characteristic path length, small-worldness, global efficiency, and rich club coefficient showed statistically significant differences between control and dementia patient groups.

Thus, these graph theory measures could potentially be used as quantitative biomarkers in cognitive impairment and dementia diagnostics.

Further studies with a larger patient cohort should evaluate and validate the diagnostic certainty and prognostic value of these graph theory measures.

## Figures and Tables

**Figure 1 brainsci-11-00943-f001:**
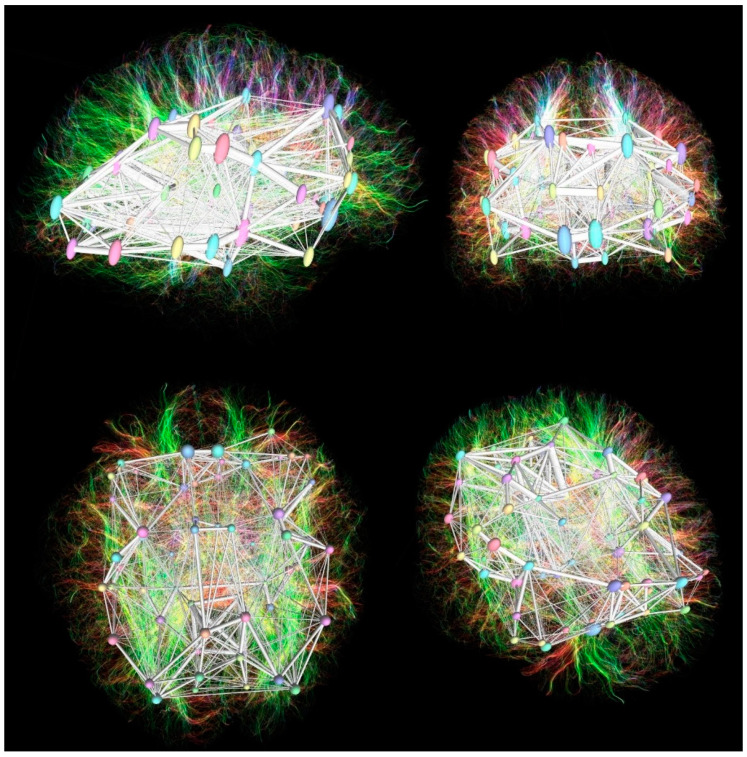
The control group patients’ DTI tractography with overlayed DKT cortical parcellation regions and connectivity matrix reconstruction in sagittal (**upper left**), coronal (**upper right**), axial (**lower left**), and oblique (**lower right**) projections.

**Table 1 brainsci-11-00943-t001:** Patient age and MoCA scores in study groups (D—dementia group, C—control group).

	MoCA		Age	
	D	C	D	C
Participants	11	11	11	11
Mean	12.818	26.182	75.000	62.273
Median	13	25	71	69
Std. Deviation	5.036	2.750	10.488	15.395
Minimum	4	23	65	35
Maximum	20	30	96	77

**Table 2 brainsci-11-00943-t002:** Mann–Whitney U test comparing control group and dementia group.

Mann–Whitney U Test		
	W	*p*
Graph Density	27.000	0.030 *
Average Clustering Coefficient	34.000	0.088
Network Characteristic Path Length	94.000	0.028 *
Small-Worldness	30.000	0.047 *
Global Efficiency	26.000	0.023 *
Assortativity Coefficient	69.000	0.606
Rich Club (k = 5)	27.000	0.030 *
Rich Club (k = 10)	29.000	0.042 *
Rich Club (k = 15)	27.000	0.028 *
Rich Club (k = 20)	30.000	0.047 *

* *p* < 0.05.

**Table 3 brainsci-11-00943-t003:** Graph density descriptive statistics in dementia group (D) and control group (C).

	Graph Density	
	D	C
Mean	0.621	0.679
Std. Error of Mean	0.017	0.018
Median	0.625	0.687
Std. Deviation	0.057	0.059
Minimum	0.537	0.563
Maximum	0.689	0.753

**Table 4 brainsci-11-00943-t004:** Average clustering coefficient descriptive statistics in dementia group (D) and control group (C).

	Average Clustering Coefficient	
	D	C
Mean	0.807	0.825
Std. Error of Mean	0.008	0.011
Median	0.802	0.832
Std. Deviation	0.027	0.036
Minimum	0.759	0.739
Maximum	0.841	0.860

**Table 5 brainsci-11-00943-t005:** Network characteristic path length descriptive statistics in dementia group (D) and control group (C).

	Network Characteristic Path Length	
	D	C
Mean	1.393	1.334
Std. Error of Mean	0.018	0.018
Median	1.388	1.324
Std. Deviation	0.059	0.059
Minimum	1.330	1.260
Maximum	1.483	1.448

**Table 6 brainsci-11-00943-t006:** Small-worldness descriptive statistics in dementia group (D) and control group (C).

	Small-Worldness	
	D	C
Mean	0.580	0.620
Std. Error of Mean	0.012	0.016
Median	0.589	0.624
Std. Deviation	0.041	0.051
Minimum	0.511	0.510
Maximum	0.633	0.682

**Table 7 brainsci-11-00943-t007:** Global efficiency descriptive statistics in dementia group (D) and control group (C).

	Global Efficiency	
	D	C
Mean	0.797	0.826
Std. Error of Mean	0.009	0.009
Median	0.798	0.830
Std. Deviation	0.029	0.029
Minimum	0.754	0.769
Maximum	0.830	0.862

**Table 8 brainsci-11-00943-t008:** Rich club descriptive statistics with different k values in the dementia group (D) and control group (C).

	Rich Club(k = 5)		Rich Club(k = 10)		Rich Club(k = 15)		Rich Club(k = 20)	
	D	C	D	C	D	C	D	C
Mean	0.621	0.679	0.628	0.685	0.657	0.702	0.704	0.736
Std. Error of Mean	0.017	0.018	0.017	0.017	0.014	0.016	0.011	0.015
Median	0.625	0.687	0.625	0.687	0.677	0.714	0.702	0.753
Std. Deviation	0.057	0.059	0.057	0.058	0.046	0.054	0.035	0.051
Minimum	0.537	0.563	0.538	0.563	0.558	0.597	0.656	0.619
Maximum	0.689	0.753	0.689	0.753	0.705	0.753	0.764	0.796

## Data Availability

The data presented in this study are available on request from the corresponding author. The data are not publicly available due to data privacy.

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
