# Peer review of "Brain Structural Connectivity Differences in Patients with Normal Cognition and Cognitive Impairment"

_brainsci, 2021, doi:10.3390/brainsci11070943_

Round 1

Reviewer 1 Report

The topic is of gret interest nowadays. Your work is well written and conducted, but I have a few comments for you.

  • I would ask for a better description of the criteria for dividing the two groups (demented vs non demented). At line 81 you briefly say that the division was based on neurological assessment and MoCA and you make a hint on functional autonomy . Do we have scales? Scores' cut-off?
  • You didn't perform a traditional morphometric brain MRI? Or you didn't mention it?

Author Response

Dear reviewer,

Thank You for Your time and review!

We have added the criteria on how we selected patients. The selection was done by focusing on MoCA test results.

Our MoCA cut-off values for the control group were ≥23 and for the dementia group ≤22. These values can produce high sensitivity, specificity, and AUC.  

In this research, our main goal was to analyze DTI connectivity patterns. So we did not analyze the morphometric features of the subjects in this study.

Thank You!

Best regards,

Nauris Zdanovskis

Reviewer 2 Report

Dear authors,

Dementia is a huge scientific and research topic. Researchers tend to identify biomarkers in earlier stages of the disease.

Unfortunately, I have to mention few limitations of your study

The sample size is too small. We have many studies assessing functional connectivity in neurocognitive disorders, we need to enlarge the samples to drive us from group to individual results. There is no DTI connectivity analysis in Dementia to my knowledge is that the first attempt? If so, you need to mention it.

The groups differ in age distribution significantly, and moreover, there is no data regarding their educational level. MoCA is a sensitive test for MCI but does not assess functionality or emotional state at all. A thorough neurological examination needs to be performed and presented.

According to which criteria dementia was diagnosed, and it was all subjects with Alzheimer's Disease or other dementia types as well?

How did you choose the metrics that you will analyze regarding DTI connectivity?

What is the added value of DTI compared to the already numerous results of the functional connectivity?  How can that be explained based on the disease pathophysiology?

It would be also interesting to apply the same metrics in MCI or SCI groups. As I mentioned earlier it is important to identify sensitive biomarkers at early stages and not when MoCA is already 4.

Few figures presenting the networks would add to the manuscript presentation.

Author Response

Dear reviewer,

Thank You for Your time, review, and comments. I will try to address comments one by one.

  1. Regarding sample size and age distribution,

Completely agree that the sample size is too small to draw definite conclusions. This was an exploratory study (rather than confirmatory) to identify whether there are differences in-between groups. In our study, we found that even with a small sample size we can see some statistically significant differences.

We plan to expand our study group in the future, including more individuals in the control group and dementia group. We mention it in the discussion section at the limitations of the study.

  1. MoCA cutoff values and education level.

In our study design For the control group, we used MoCA cutoff scores of  ≥23, and for the dementia group, ≤22 as these values can produce high sensitivity, specificity, and AUC.

Also, we added information regarding education level. All of the participants in the control group and dementia group had at least 16 years of education.

  1. Functional connectivity and structural connectivity.

Functional connectivity and structural connectivity go hand-in-hand.

Structural connectivity represents fiber pathways and functional connectivity represents “information flow”. Functional connectivity will be present if the fiber pathways are there.

There are studies that explore topological changes in structural and functional networks (for example, Wang et al. https://www.frontiersin.org/articles/10.3389/fnagi.2018.00404).

So, our idea is that there may be connectivity pattern changes in prodromal cognitive impairment and in patients who are at higher risk of developing cognitive impairment.

  1. Were all subjects with Alzheimer's Disease or other dementia types as well?

All subjects included in our study had mixed-type dementia.

  1. How did you choose the metrics that you will analyze regarding DTI connectivity?

We used connectivity measures that can be found in the “Brain Connectivity Toolbox” where the main contributors are Olaf Sporns, Mikail Rubinov, Yusuje Adachi et al. (https://sites.google.com/site/bctnet/Home/help).

While there are many more measures that could be used, we focused on metrics that are already somewhat researched and analyzed in other studies.

  1. Added-value of DTI compared and disease pathophysiology.

DTI connectivity analysis added value could be additional biomarkers.

Based on pathophysiology brain structural architecture could identify individuals at risk, i.e., if the patient has lower “global efficiency” or lower “rich-club” coefficient it could show us that the individual has a smaller amount of structural cognitive reserve. A similar analogy we see, for example, in stroke treatment and diagnostics – if the patient has good collaterals - the better therapy results are.

Also, DTI examination is faster (from 5 to 8 minutes) than fMRI (usually from 10 to 15 minutes) and DTI is not so prone to the artifacts comparing to the fMRI scanning.

  1. Early-stage sensitive biomarkers.

This is our first step to identify these DTI connectivity biomarkers. Further, we plan to do the follow-up for the patients with normal cognition and cognitive impairment and analyze longitudinal data.

Of course, when we will have a larger patient group we plan to utilize these DTI connectivity measures in patients with MCI and SCI too.

Also, we added a figure that shows DTI tractography with cortical parcellation regions and connectivity matrix reconstruction. 

Thank You!

Best regards,

Nauris Zdanovskis